# JenBridge: Adaptive Long-Form Video Sound-tracking across Scene Transitions

## Abstract

We address the challenge of generating high-fidelity, long-form soundtracks that remain coherent across scene transitions. Existing AI music systems are mainly designed for short, isolated clips and lack mechanisms to ensure narrative continuity. We present `JenBridge`, a modular and interpretable framework for adaptive long-form video soundtracking that ensures both high-fidelity audio generation and transition naturalness. The core architecture is a Transformer-based generative model trained with a flow-matching objective, following a two-stage paradigm: pretraining on large-scale text–audio corpora to establish robust musical priors, then adapting to the video domain with dual text–visual conditioning for precise cross-modal alignment. Crucially, to achieve long-form coherence across diverse scene changes, `JenBridge` incorporates a novel adaptive transition mechanism. This system features a versatile toolkit of transition styles, including a generative transition method, and uniquely employs a Large Language Model (LLM) Agent that acts as a director to select the most appropriate transition for each narrative shift intelligently. To rigorously assess this task, we propose the LVS Benchmark, a new benchmark that includes a curated dataset and novel evaluation metrics focusing on holistic and transition-aware assessment. Extensive experiments on the proposed benchmark demonstrate that `JenBridge` significantly outperforms existing methods in both objective and subjective metrics, particularly in terms of transition naturalness and overall narrative coherence. JenBridge represents a significant step towards fully automated, professional-quality video soundtracking. The codes and benchmark will be made publicly available.

## 1 Introduction

The ability to automatically generate high-fidelity music stands as a significant milestone in the development of creative AI systems. This progress is largely driven by breakthroughs in text-to-music (T2M) synthesis, where frontier models Copet et al. (2023); Suno AI (2025); Bai et al. (2024) have set a new standard for audio fidelity and coherence. These powerful generative capabilities have inspired growing interest in the more complex, cross-modal task of video-to-music (V2M) generation. The predominant paradigm involves analyzing visual semantics Di et al. (2021); Tian et al. (2025b); Zuo et al. (2025) or motion tracks Yu et al. (2023); Zhu et al. (2022b); Su et al. (2024) from a short video clip to generate a single, corresponding piece of music. However, this paradigm suffers from critical limitations for practical use. First, the generation pipeline is typically unmodifiable, offering users little creative control over intermediate steps or the final output. Second, these models lack the mechanisms to handle scene transitions, confining them to generating monolithic audio for single clips. This deficiency renders them impractical for real-world, long-form content that is characterized by dynamic scene changes.

To address these limitations, we introduce `JenBridge`, a modular and interpretable framework for adaptive long-form video soundtracking. Our framework decomposes this complex task by first segmenting the input video into a sequence of semantically coherent clips. The subsequent process consists of two core components: per-segment music generation and inter-segment adaptive transition. The first component is a powerful video-aware generative model responsible for per-segment music synthesis, built and trained from scratch. It incorporates a pre-trained neural audio codec Défossez et al. (2022) to encode raw waveforms into a compact and expressive latent representation. The generative backbone that operates in this latent space is a Multimodal Diffusion

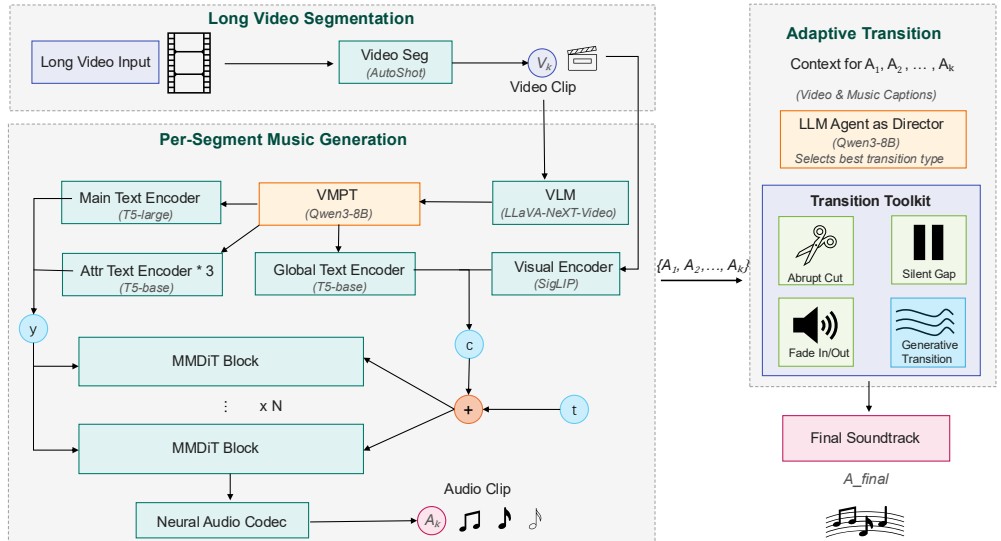

Figure 1: An overview of the `JenBridge` framework. Our framework consists of three main parts: (1) A video segmentation module that partitions a long video into clips ($V_k$). (2) A per-segment music generation module that synthesizes a corresponding audio clip ($A_k$) using a video-aware generative model. The model is guided by a sequence-level condition ($y$) from text encoders and a global condition ($c$) that fuses pooled text embeddings with visual features. (3) An adaptive transition module where an LLM Agent selects the optimal transition style from a versatile toolkit to coherently connect the audio clips into a final soundtrack ($A_{\text{final}}$).

Transformer (MMDiT)Esser et al. (2024). Trained with a flow-matching objectiveLipman et al. (2022), this architecture achieves both high-fidelity synthesis and efficient inference. To enable precise cross-modal alignment, we extend the generative architecture to the video domain with a dual conditioning scheme that combines a fine-tuned multi-encoder text architecture for semantic control and a dedicated visual encoder. The second component is a novel adaptive transition mechanism designed to seamlessly connect individual clips. As illustrated in Figure 1, this mechanism provides a versatile toolkit of transition styles, including a ControlNet-based Zhang et al. (2023) generative transition model, and uniquely employs an LLM agent acting as a creative director to select the most appropriate transition for each narrative shift.

Furthermore, to address the critical gap in evaluation methodologies, we introduce the Long-form Video Soundtrack (LVS) Benchmark. As the first benchmark specifically designed for long-form video soundtracking, LVS provides a curated set of multi-scene movie trailers with rich, fine-grained multimodal annotations, including music and video captions as well as scene transition points. Besides, the LVS Benchmark proposes a new evaluation paradigm that emphasizes holistic, long-form assessment and introduces procedures for evaluating the quality and appropriateness of complete soundtracks with transitions, moving beyond simple segment-based metrics.

In conclusion, our main contributions are three-fold:

- We present **JenBridge**, an end-to-end framework for long-form video soundtracking that produces coherent, high-fidelity music across diverse scene transitions. Its modular architecture is inherently interpretable and controllable, positioning it as a practical tool for creative workflows.

- We propose a novel **adaptive transition mechanism**, which combines a versatile toolkit of transition styles, including a generative transition method, with an LLM Agent that acts as a director to make context-aware creative choices.

- We introduce the **LVS Benchmark**, a comprehensive benchmark with rich annotations and a holistic evaluation protocol for long-form video soundtracking. On this benchmark, `JenBridge` achieves state-of-the-art performance, significantly advancing music quality, video–music alignment, and transition naturalness.

## 2 RELATED WORKS

### 2.1 TEXT-TO-MUSIC GENERATION

Text-to-music generation has recently seen an explosion of progress, largely driven by advancements in generative architectures. Initial breakthroughs were made by autoregressive models like MusicLM Agostinelli et al. (2023) and MusicGen Copet et al. (2023), which excel at generating musically structured audio but are constrained by slow, sequential inference. Subsequently, diffusion models became prominent, significantly improving audio fidelity in works such as Noise2Music Huang et al. (2023), ERNIE-Music Zhu et al. (2023a), MusicLDM Chen et al. (2024), Mousai Schneider et al. (2024), and Jen-1 Li et al. (2024a), with open-source contributions like Stable Audio Open Evans et al. (2025) further democratizing access. To address the efficiency challenges of these paradigms, recent work has explored alternative paths, such as non-autoregressive Transformers Ziv et al. (2024). The emergence of rectified flow Lipman et al. (2022) represents the latest advancement, offering a compelling balance of high-fidelity synthesis and rapid, few-step inference, as demonstrated by models like MusicFlow Prajwal et al. (2024). Our work builds upon this state-of-the-art approach, utilizing a Transformer-based architecture with the flow-matching training objective as our foundation. Beyond core generation, the field is also pushing towards unified frameworks for high-quality and controllable generation Bai et al. (2024); Melechovsky et al. (2023), greater user interactivity Nistal et al. (2024); Yao et al. (2025), advanced reasoning with chain-of-thought prompting Lam et al. (2025), and the challenging task of long-form generation Yuan et al. (2025). This vibrant academic landscape is mirrored by the rapid development of widely-used commercial systems Suno AI (2025); Udio (2025); ElevenLabs (2024); Producer AI (2025).

### 2.2 VIDEO-TO-MUSIC GENERATION

Building upon the advances in the high-fidelity text-to-music synthesis domain, video-to-music generation introduces a significant additional challenge of ensuring robust audio-visual alignment. Many approaches tackle this by employing hierarchical visual features, using high-level semantics for mood and melody Gan et al. (2020); Kang et al. (2024); Di et al. (2021); Li et al. (2024c) while leveraging low-level motion cues for rhythm Li et al. (2024d); Liu et al. (2025); Qi et al. (2025). Others focus on advanced modeling for alignment, exploring attention mechanisms Zuo et al. (2025); Lin et al. (2025); Li et al. (2024b). Recent trends also include scaling up with massive "in-the-wild" datasets Su et al. (2024); Zhou et al. (2025); Chi et al. (2024); Tian et al. (2025b) and developing unified any-to-audio models Liu et al. (2024); Tian et al. (2025a).

To achieve tighter temporal synchronization, some research narrows the focus to specific domains with strong motion-music correlations, such as dance Liang et al. (2024); Zhu et al. (2022a); Sun et al. (2025); Zhu et al. (2022b) or other rhythmic activities like sports Yu et al. (2023); You et al. (2024). Concurrently, another line of work leverages the high-level reasoning capabilities of Large Language Models (LLMs) to simulate complex creative workflows Xie et al. (2025) or to bridge modalities without paired data via chain-of-thought reasoning Guan et al. (2025).

Despite this progress, a critical limitation remains: existing methods treat videos as monolithic segments, generating a single piece of music and largely overlooking the challenge of creating adaptive and musically coherent **transitions** between semantically distinct scenes. Our work, `JenBridge`, is specifically designed to address this crucial gap.

## 3 METHODOLOGY

Our framework, `JenBridge`, generates a long-form musical piece $A_{\text{final}}$ for an arbitrary-length video. The process can be conceptually divided into three main stages: (1) video segmentation, (2) per-segment music generation via a progressive, video-aware model, and (3) adaptive transitions to connect the music segments. We describe each stage in detail below.

### 3.1 Semantic Video Segmentation

The process begins by partitioning the input video $V_{\text{long}}$ into a temporally ordered set of $K$ semantically coherent clips, $\{V_1, V_2, \ldots, V_K\}$ using PySceneDetect Castellano (2020). This preprocessing step allows our framework to handle videos of any duration by operating on manageable segments.

### 3.2 Progressive Training for Video-Aware Music Generation

Our video soundtracking model is developed through a progressive, two-stage training paradigm. We first built a foundational text-to-music model with a pre-trained text conditioning architecture. Subsequently, we adapt this model to be video-aware by introducing a direct visual conditioning signal and a mechanism to generate text prompts from video content.

**Audio Representation and Core Architecture**  The first stage constructs a foundational model for high-fidelity music generation, beginning with a pre-trained neural audio codec Défossez et al. (2022) that encodes 48 kHz stereo audio into a compact, expressive latent representation, $L$. This latent space is modeled by a Multimodal Diffusion Transformer (MMDiT) Esser et al. (2024), which is designed to accept two distinct types of conditioning signals: a detailed sequence embedding $y$ and a global pooled embedding $c$.

**Text Conditioning Architecture**  The conditioning mechanism for the foundational model is based on a fine-tuned multi-encoder text architecture that produces both the sequence and pooled embeddings from text prompts. For a given clip $V_k$ with text prompts, we generate:

1. **A sequence embedding** $y_{\text{text},k}$, which provides rich, token-level semantic information. It is constructed from the main descriptive prompt $(P_{k,d})$ and three attribute prompts $(P_{k,attr_i,i=1,2,3})$: genre, instrument, and mood. A T5-large Raffel et al. (2020) encoder computes the main embedding $E_d = \mathcal{E}_{\text{T5-L}}(P_{k,d})$, while three fine-tuned T5-base Raffel et al. (2020) encoders compute attribute embeddings $E_{attr_i,i=1,2,3}$. These are concatenated to form the final sequence condition:

$$y_{\text{text},k} = \text{concat}_{\text{seq}}(\text{concat}_{\text{dim}}(E_{attr}), \text{pad}(E_d)) \tag{1}$$

2. **A global text embedding** $c_{\text{text},k}$, which captures the overall essence of the prompt. It is derived by encoding the main prompt with a fine-tuned T5-base encoder, $\mathcal{E}_{\text{T5-B}}$, and taking its pooled output:

$$c_{\text{text},k} = \text{pool}(\mathcal{E}_{\text{T5-B}}(P_{k,d})) \tag{2}$$

**Video Conditioning and Fusion**  In the second stage, we introduce a direct visual signal. For each video clip $V_k = \{f_1, \ldots, f_T\}$, where $f_i$ denotes the $i$-th frame, we use SigLIP Zhai et al. (2023) as the visual encoder $\mathcal{E}_V$ to extract frame-level features. These frame-level features are then average-pooled across the temporal dimension to obtain a global visual feature vector $F_{k,v}$:

$$F_{k,v} = \frac{1}{T} \sum_{t=1}^{T} \mathcal{E}_{\text{V}}(f_t) \tag{3}$$

This visual feature is then concatenated with the global text embedding $c_{\text{text},k}$ from the previous step to form the final, comprehensive pooled condition $c_{\text{fused},k}$:

$$c_{\text{fused},k} = \text{concat}(c_{\text{text},k}, F_{k,v}) \tag{4}$$

**From Video Captions to Music Prompts (VMPT)**  Training the video-aware model in the second stage requires the same text prompt patterns $P_k$ that are originally absent in the vanilla video-music dataset. To this end, we introduce the Visual-to-Music Prompt Translator (VMPT), a fine-tuned Qwen3-8B Yang et al. (2025) LLM $\mathcal{F}_{\text{VMPT}}$ to transcribe a raw video caption $C_k$ from a pre-trained captioner into a structured music prompt $P_k$ that includes several music metadata like genre, instrument, mood, and BPM:

$$P_k = \mathcal{F}_{\text{VMPT}}(C_k) \tag{5}$$

We train VMPT separately on a carefully curated collection of music–video caption pairs. Since we use the same model to generate these captions during inference and text-to-music training, the trained VMPT module can be directly plugged into the inference pipeline.

**Training Objective and Generation**   Both stages of our progressive training paradigm are optimized using the same conditional flow-matching objective Lipman et al. (2022). The model's network, $v_\theta$, learns to predict a ground-truth vector field $u_t(x_1, z)$ based on the text sequence condition $y_{\text{text},k}$ and a global condition $c$. The training loss is defined as:

$$\mathcal{L}_{\text{FM}}(\theta) = \mathbb{E}_{t,x_1,z,y_k,c} \left[ \|v_\theta(x_t, t, y_{\text{text},k}, c) - u_t(x_1, z)\|^2 \right] \tag{6}$$

Here, $x_t$ is a point on the probability path between a data sample $x_1$ and a noise sample $z$, and $t \in [0, 1]$. The global condition $c$ is stage-dependent: for stage 1 (text-to-music), we use the text-only global embedding $c = c_{\text{text},k}$, while for stage 2 (video-aware adaptation), it is updated to the fused signal $c = c_{\text{fused},k}$, which incorporates visual information.

### 3.3   ADAPTIVE MUSIC TRANSITION

Generating the set of audio clips $\{A_1, \ldots, A_K\}$ is only part of the challenge; connecting them seamlessly is crucial for a coherent long-form soundtrack. Our framework employs an adaptive mechanism that intelligently selects the most suitable transition style between any two adjacent clips, $A_k$ and $A_{k+1}$. This involves a versatile *Transition Toolkit* and a directing *LLM Agent* that acts as a *director*, choosing the appropriate protocol for each scene change.

**The Transition Toolkit**   To handle diverse narrative requirements, our framework categorizes the connection between two music clips into four distinct styles, managed by the toolkit. These include an `abrupt cut` for sharp scene changes, a `silent gap` for creating dramatic pauses, and a `fade-out/fade-in` for gentle, gradual shifts. The most sophisticated style is a `generative transition`, which synthesizes a novel musical bridge to contextually and seamlessly link two distinct audio segments.

Each style is realized through a specific technique. The `abrupt cut` and `silent gap` are implemented via direct concatenation, while `fade-out/fade-in` is achieved with linear volume modulation. The `generative transition` is powered by a specialized inpainting model, a ControlNet-based Zhang et al. (2023) adaptation of our text-to-music model. We opt for this inpainting-based strategy due to the scarcity of dedicated musical transition data. An inpainting model can be effectively trained on our abundant text-audio corpora by simply masking segments of the audio, thereby learning to fill gaps conditioned on context. This approach allows us to create a robust inpainting model that can then be adapted to the transition purpose in a training-free manner. During inference, this model is guided by a condition produced via a two-step interpolation process. Both steps employ spherical linear interpolation (slerp), which is preferred for high-dimensional embeddings as it maintains vector magnitude and follows a geodesic path on the hypersphere, often leading to more perceptually uniform transitions. Given two vectors $v_1, v_2$ and a parameter $\tau \in [0, 1]$ to control the interpolation threshold, slerp is defined as:

$$\text{slerp}(v_1, v_2, \tau) = \frac{\sin((1 - \tau)\theta)}{\sin\theta} v_1 + \frac{\sin(\tau\theta)}{\sin\theta} v_2 \tag{7}$$

where $\theta = \arccos\left(\frac{v_1 \cdot v_2}{\|v_1\|\|v_2\|}\right)$ is the angle between the vectors. The two interpolation steps are as follows:

1. **Text Embedding Interpolation:** The text embeddings $E_k$ and $E_{k+1}$ of the adjacent clips are spherically interpolated to create a smooth semantic transition:

$$E_{\text{interp}} = \text{slerp}(E_k, E_{k+1}, \lambda) \tag{8}$$

   where we set the interpolation factor $\lambda = 0.5$.

2. **Block-wise Latent Interpolation:** The latent representations $L_k, L_{k+1} \in \mathbb{R}^{D \times T_L}$ are partitioned into $N$ blocks. A fused boundary condition is created by applying block-wise slerp:

$$L'_{\text{k,i}} = \text{slerp}(L_{k,i}, L_{k+1,N-i+1}, \alpha_i), \quad \text{for } i = 1, \ldots, N \tag{9}$$

   where $L_{k,i}$ is the $i$-th block of $L_k$, $L_{k+1,N-i+1}$ is the corresponding block from the end of $L_{k+1}$, and $\alpha_i$ is a small, progressively increasing weight from 0.1 to 0.5.

This combined interpolated condition then guides the inpainting model to synthesize a coherent musical bridge.

**LLM Agent as Transition Director**  To intelligently select the appropriate technique from the transition toolkit, we employ an LLM Agent (Qwen3-8B Yang et al. (2025)) that functions as a creative director. For each transition point between two video clips, the agent is provided with comprehensive contextual information: the descriptive visual caption and the generated musical prompt for both the preceding and succeeding segments. This decision-making is managed via few-shot in-context learning, where the agent's prompt includes this context alongside a set of curated examples of ideal transition choices for different narrative scenarios. Based on this information, the LLM selects and outputs the name of the most suitable technique from the toolkit (e.g., `generative transition`, `abrupt cut`) to apply.

## 4  EXPERIMENTS

### 4.1  TRAINING SETUP

#### 4.1.1  TRAINING THE VISUAL-TO-MUSIC PROMPT TRANSLATOR (VMPT)

We train the VMPT on the V2M-finetuning dataset (V2M-20k) introduced by Tian et al. (2025b). For each video clip, a descriptive video caption is generated using the LLaVA-NeXT-Video-DPO-7B model Zhang et al. (2024), while music tags are extracted with a proprietary pre-trained tagging model. The tags are transformed into diverse, high-quality music captions through a rule-based system, yielding 18k caption pairs. A Qwen3-8B LLM Yang et al. (2025) is then fine-tuned on this corpus with LoRA Hu et al. (2022) using the LLaMAFactory framework Zheng et al. (2024). Training is conducted for 3 epochs on two NVIDIA A800 80G GPUs.

#### 4.1.2  STAGE 1: FOUNDATIONAL TEXT-TO-MUSIC MODEL TRAINING

We train the main text-to-music generative model for 400k steps on 64 NVIDIA A800 GPUs using a private database of 100k licensed high-fidelity songs, totaling 3,700 hours of audio. Building on this stage, the specialized inpainting model for generative transitions is fine-tuned from the corresponding checkpoint. This model is further trained for 100k steps on the same dataset with masking applied, using 16 NVIDIA A800 GPUs.

#### 4.1.3  STAGE 2: VIDEO-AWARE ADAPTATION

The video–music alignment is trained on the V2M dataset Tian et al. (2025b). We filter a subset of 110k samples and truncate video clips to a maximum length of 30 seconds, yielding 600 hours of video–music pairs in total. The model is trained for 200k steps on 8 NVIDIA A800 GPUs.

### 4.2  A COMPREHENSIVE BENCHMARK FOR LONG-FORM VIDEO SOUNDTRACKING

Due to the focus of existing work on short video clips, there is currently a lack of an available benchmark for comprehensively evaluating long-form video soundtracking, especially regarding scene changes and transition quality. To this end, we propose the **Long-form Video Soundtrack (LVS) Benchmark**. The LVS benchmark comprises 120 long-form trailer videos manually filtered from the large-scale MMTrailer Chi et al. (2024) dataset. Our selection process prioritizes videos with rich narrative variation and a high density of distinct scene transitions. The benchmark totals 3 hours of footage and contains 567 different scenes, with an average of 4.72 semantic segments requiring musical transitions per video. All videos were processed with our segmentation module to generate scene boundary annotations, and each segment has a corresponding visual description generated by a pre-trained captioner Zhang et al. (2024).

It should be noted that we do not provide the original corresponding music, and some of them do not even have original musical soundtracks. We believe video soundtracking is an inherently subjective creative endeavor, and different artists will invariably produce distinct musical interpretations for the same visual content. We therefore argue that there is no "ground-truth" soundtrack, and the similarity to the original music should not be a metric for quality. The metrics we introduce below also do not include any direct comparison to their original audio tracks. Our benchmark is designed not to calculate the musical similarity, but to serve as an open-ended benchmark for evaluating the quality of generated soundtracks based on their alignment with the visual narrative.

### 4.2.1 BASELINES

We select four state-of-the-art, open-source models as our main baselines: CMT Di et al. (2021), LORIS Yu et al. (2023), AudioX Tian et al. (2025a), and VidMuse Tian et al. (2025b). For a fair comparison on the long-form task, the default approach is to generate music for each video segment independently and then concatenate the results. Since both CMT and VidMuse are also capable of generating long-form tracks, we evaluate them in two settings: the concatenated short-clip version (denoted as $CMT_S$ and $VidMuse_S$) and a version where the entire long video is processed directly (denoted as $CMT_L$ and $VidMuse_L$), resulting in a total of six baseline approaches for comparison.

### 4.2.2 OBJECTIVE METRICS

To quantitatively evaluate our model, we employ a series of objective metrics assessing music quality and video-music alignment. Since our LVS Benchmark does not contain ground-truth soundtracks, we utilize reference-free and cross-modal metrics. To measure local video-music alignment, we use the **ImageBind Score** Girdhar et al. (2023), calculated as the average cosine similarity between the embeddings of each video segment and its corresponding generated audio clip. For a more nuanced evaluation of musicality, we adopt three axes from the Meta Audiobox Aesthetics framework Tjandra et al. (2025), all computed on the full-length audio: **Production Quality (PQ)**, which assesses technical aspects such as clarity and dynamics; **Production Complexity (PC)**, which measures the richness of the audio scene and transition; and **Content Enjoyment (CE)**, which captures subjective artistic quality. It is important to note that the Audiobox Aesthetics scores are computed holistically on the entire long-form soundtrack, ensuring that the quality of musical transitions significantly impacts the final results.

### 4.2.3 SUBJECTIVE METRICS (USER STUDY)

As soundtrack quality evaluation can be very subjective, we conduct a comprehensive user study with 10 participants from diverse backgrounds. In a randomized and blind setting, they evaluated soundtracks generated by our model and baselines for a variety of videos. Participants rated each soundtrack on a 5-point Likert scale (1=Poor, 5=Excellent) across three key dimensions: **Music Quality** (clarity, richness), **Video-Music Alignment** (correspondence of the music's mood and rhythm to the visuals), and **Transition Naturalness** (the smoothness and appropriateness of musical changes between scenes), as well as their average score as the overall performance.

### 4.3 MAIN RESULTS

We present our main experimental results in Table 1, where we compare `JenBridge` against six baseline approaches on our proposed LVS Benchmark. The evaluation encompasses a comprehensive set of both objective and subjective metrics, designed to assess everything from audio quality and cross-modal alignment to transition coherence and user preference.

Table 1: Comprehensive comparison with baseline models on the LVS Benchmark. For all metrics, higher is better. The best results are highlighted in **bold**.

| Model | Objective Evaluation | | | | Subjective Evaluation (User Study) | | | |
|---|---|---|---|---|---|---|---|---|
| | ImageBind$_{avg}$ ↑ | PQ ↑ | PC ↑ | CE ↑ | Music ↑ | Alignment ↑ | Transition ↑ | Overall ↑ |
| $CMT_S$ | 0.143 | 5.52 | 4.21 | 5.38 | 3.3 | 3.2 | 1.6 | 2.7 |
| $CMT_L$ | 0.115 | 5.61 | 4.88 | 5.45 | 3.4 | 3.0 | 2.0 | 2.8 |
| LORIS | 0.121 | 4.81 | 4.15 | 4.52 | 2.9 | 2.7 | 1.6 | 2.4 |
| AudioX | 0.132 | 6.55 | 4.35 | 6.41 | 3.8 | 3.7 | 1.7 | 3.1 |
| $VidMuse_S$ | 0.162 | 6.81 | 4.42 | 6.75 | 3.8 | 3.8 | 1.8 | 3.1 |
| $VidMuse_L$ | 0.148 | 6.89 | 5.56 | 6.82 | 3.9 | 3.8 | 2.5 | 3.4 |
| **JenBridge (Ours)** | **0.224** | **8.12** | **7.83** | **8.21** | **4.4** | **4.3** | **4.2** | **4.3** |

**Objective Results.** The objective results, presented in Table 1, highlight the comprehensive advantages of our approach. In terms of video-music alignment, `JenBridge` achieves an ImageBind Score that is over 38% higher than the strongest baseline, $VidMuse_S$. This underscores the difficulty that existing long-form generation models face in maintaining semantic alignment over extended

Table 2: Ablation studies on the components of our framework. We report key objective and subjective metrics to assess the impact of each component.

| Model Configuration | PQ ↑ | ImageBind$_{avg}$ ↑ | Transition ↑ |
|---|---|---|---|
| **JenBridge (Full Model)** | **8.12** | **0.224** | **4.2** |
| *Core Contributions:* | | | |
| w/o Adaptive Transition (Abrupt Cut) | 7.89 | 0.195 | 2.8 |
| w/o Visual Conditioning (Text-Only) | 7.65 | 0.171 | 3.1 |
| *Key Components:* | | | |
| w/o VMPT (Raw Video Caption) | 7.48 | 0.185 | 3.4 |
| w/o LLM Agent (Always Generative) | 7.91 | 0.221 | 3.5 |
| w/o Slerp in Transition (Use Lerp) | 8.04 | 0.219 | 3.9 |

durations, as evidenced by the performance drop in their respective long-form variants ($CMT_L$ and $VidMuse_L$). Furthermore, our model leads in all audio aesthetics axes. The most significant advantage is observed in Production Complexity (PC), where `JenBridge` outperforms the next best long-form baseline by 2.27. This objectively validates that our adaptive transition mechanism creates a richer and more complex musical structure compared to the simple concatenation or monolithic generation of baselines, whose PC scores remain low.

**Subjective Results.**   The user study results strongly corroborate our objective findings and further emphasize the perceptual superiority of our method. While participants rated `JenBridge` significantly higher than all baselines on both music quality and alignment, the most pronounced difference is in the transition score. Our model's score is 0.9 higher than the best-performing baseline, $VidMuse_L$, and more than doubles the scores of all concatenation-based methods. This clear superiority in handling scene changes directly validates the effectiveness of our adaptive transition mechanism and translates to the highest Overall score, confirming that users perceive the soundtracks generated by `JenBridge` as the most coherent and compelling.

## 4.4 ABLATION STUDIES

To validate the effectiveness of each key component and design choice within our `JenBridge` framework, we conduct a comprehensive series of ablation studies. We systematically remove or replace individual modules from our full model and evaluate the impact on performance. The results, summarized in Table 2, demonstrate the contribution of each component.

**Analysis of Ablation Results.**   The ablation study results in Table 2 clearly demonstrate the importance of each component in our framework. Removing the entire Adaptive Transition mechanism and reverting to simple concatenation causes a catastrophic drop in the subjective Transition score from 4.2 to 2.8. This highlights that the adaptive transition strategy is critical for long-form soundtracking. Disabling visual conditioning leads to a sharp decline in the ImageBind score from 0.224 to 0.171, confirming that direct visual features provide essential cues for alignment. Using raw video captions without the VMPT also results in a general degradation across all metrics, validating its effectiveness in translating visual concepts into musically potent prompts.

Within the transition mechanism, each design choice proves to be crucial. Replacing the intelligent LLM Agent with a fixed strategy severely impacts the Transition score, reducing it to 3.5, as it fails to apply the appropriate transition style for different narrative contexts. Finally, substituting slerp with a simpler linear interpolation during the generative transition process also leads to a slight decrease in performance, demonstrating the effectiveness of our advanced interpolation method for achieving smoother transitions.

## 4.5 QUALITATIVE ANALYSIS

To provide an intuitive understanding of our framework's capabilities, we present three representative case studies in Figure 2, illustrating how the LLM Agent intelligently directs the soundtrack

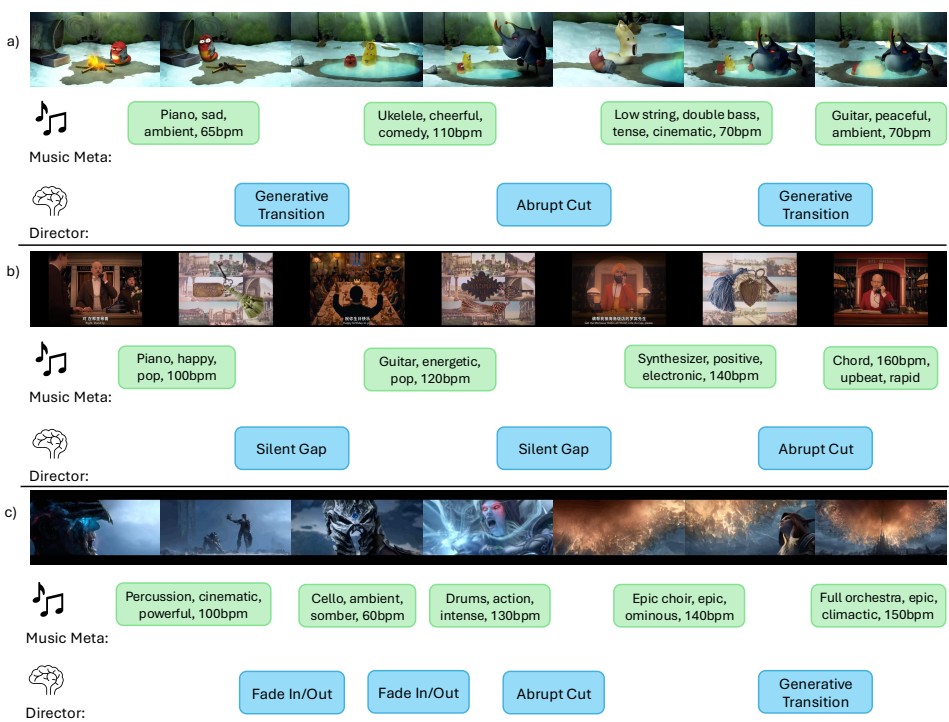

Figure 2: Qualitative examples demonstrating the LLM Agent's adaptive transition choices based on the visual content and music metadata. The top row (a) shows an animated video where the agent matches the emotional arc with varied transitions. The middle row (b) presents a cinematic montage where abrupt cuts and silent gaps enhance the drama. The bottom row (c) features an anime sequence where the transition style shifts from abrupt cut to smooth generative transition to match the changing pace.

by selecting diverse transition strategies based on the video's narrative context. For the animated video in the first row, the agent mirrors the emotional arc of the story, using a `generative transition` for a positive shift from loneliness to joy, an `abrupt cut` for a moment of sudden peril, and another `generative transition` to resolve the tension. In a cinematic trailer with disconnected scenes, the agent opts for `abrupt cuts` and `silent gaps` to enhance the dramatic impact rather than forcing a musical connection. Finally, for an anime sequence with a distinct change in pacing, the agent's strategy evolves from using `abrupt cuts` during a fast-paced action sequence to employing smoother `fade-out/fade-in` and `generative transitions` as the narrative becomes more reflective. The full generated videos for these case studies, including side-by-side comparisons with baseline methods, are provided in the supplementary materials.

## 5 CONCLUSION

In this paper, we introduced `JenBridge`, a modular and interpretable framework that successfully tackles the challenge of generating coherent, long-form soundtracks for videos with dynamic scene changes. Our approach integrates a powerful, video-aware generative model for per-segment synthesis with a novel LLM-directed adaptive transition mechanism. We also established the LVS Benchmark to facilitate rigorous evaluation, on which `JenBridge` demonstrates state-of-the-art performance, particularly in transition naturalness and overall coherence. `JenBridge` represents a significant step towards creating practical, collaborative AI tools that empower human creators, bridging the gap between automated generation and professional-quality video production. Future work includes extending the framework to incorporate sound effects and enhancing the LLM Agent with long-range narrative planning.

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

## A    STATEMENTS AND BROADER IMPACT

### A.1    ETHICS STATEMENT

**Data Usage.**    Our work adheres to strict ethical guidelines regarding data usage. The foundational text-to-music model was trained on a private, curated dataset. This dataset consists exclusively of high-fidelity, professionally produced songs for which we have secured the necessary licenses from all copyright holders. All musical content is confirmed to be non-harmful and appropriate for research. For all other training stages and for the construction of our LVS Benchmark, we utilized publicly available academic datasets, including VidMuse Tian et al. (2025b) and MMTrailer Chi et al. (2024), in accordance with their terms of use for research purposes. To respect the copyright of the original sources, our LVS Benchmark will be released as a set of annotations and processing scripts.

**User Study and Intended Use.**    Our subjective evaluations involved a user study where informed consent was obtained from all participants prior to their involvement. All collected responses were fully anonymized to protect participant privacy. We intend for `JenBridge` to serve as a collaborative tool to assist and empower, not replace, human creators. We encourage the responsible and ethical use of our publicly released codebase and benchmark.

### A.2    REPRODUCIBILITY STATEMENT

To ensure the reproducibility of our work, we will make the inference codes, the LVS Benchmark, and relevant model components publicly available. For the LVS Benchmark, we will release all our annotations, including the original video identifiers (e.g., YouTube IDs), the start and end timestamps for each segmented clip, the VLM-generated video captions, and the VMPT-generated music captions. The raw video files will not be redistributed due to copyright restrictions; however, our provided annotations will allow researchers to reconstruct the benchmark by accessing the original public sources. For the `JenBridge` model, we will release the weights for the video-aware adaptation stage (Stage 2), along with the complete training annotations used for this stage. The weights for the foundational text-to-music model (Stage 1) and its private, licensed training data will not be released. However, to facilitate further research and application, we will provide public API access to our foundational text-to-music model. The inference codebase for the v2m stage, including scripts for data processing, will also be made publicly available.

### A.3    LLM USAGE STATEMENT

In the preparation of this manuscript, we utilized a large language model (LLM) as an assistive tool. The LLM's role included improving the clarity, grammar, and style of the text. All scientific claims, experimental designs, and the core intellectual contributions presented in this paper were conceived and articulated by the human authors. The authors have reviewed, edited, and take full responsibility for all final text in this submission.

### A.4    LIMITATIONS

Despite the strong performance of `JenBridge`, we acknowledge two primary limitations in our current work. First, the musical quality of our final video-aware model is constrained by the public datasets available for video-music training. While our foundational text-to-music model is trained on a high-fidelity, licensed corpus, the public datasets used for video-aware adaptation often feature lower audio fidelity. This results in a perceptible drop in musical quality when the model is fine-tuned for the video-aware task. We believe that future performance gains can be achieved by curating larger, high-fidelity, and properly licensed video-music corpora.

Second, while our framework is a significant step forward, it is not yet a fully production-ready tool, primarily due to two factors related to its high-level contextual understanding. The first factor is the scope of its LLM Agent, which operates on a local, pairwise basis without a global, long-range plan for the entire video's narrative arc. The second is that our current model operates solely on the visual stream and does not comprehend the original audio from the video, such as human speech and background audio. This can lead to suboptimal musical choices in dialogue-heavy scenes

where the conversation dictates the emotional tone. Future work could address these limitations by incorporating the sound source separation and automatic speech recognition module to inform the LLM Agent, and by evolving the agent's capabilities towards long-range narrative planning.

# B  METHODOLOGY DETAILS

## B.1  T5-BASE ENCODER FINE-TUNING

The four specialized T5-base Raffel et al. (2020) encoders, responsible for handling attribute and global prompts, were fine-tuned from their original checkpoints to better align with musical concepts. For this process, we utilized our large-scale text-to-music training dataset. To train each specialized encoder (e.g., for genre, instrument, or mood), we created attribute-specific subsets of the data, pairing an audio clip with only its relevant attribute caption.

We fine-tuned the encoders using a self-supervised methodology following MERT Li et al. (2024e). In this setup, an audio clip is first encoded into a sequence of latent representations using Encodec Défossez et al. (2022). These latents are then processed by a 12-layer, encoder-only Transformer. The training objective is masked feature prediction: we randomly mask a portion of the audio latent sequence, and the model is tasked with reconstructing the masked content and predicting the Hubert Hsu et al. (2021) label. We train each encoder using 8 A800 GPUs for 1 epoch.

## B.2  VIDEO SEGMENTATION DETAILS

For the video segmentation stage, we employ the PySceneDetect Castellano (2020) library to partition long videos into semantically coherent clips. Specifically, to ensure that our segmentation captures only significant narrative shifts while ignoring minor camera movements or subtle visual changes, we set the detection 'threshold' parameter of the 'ContentDetector' class to 30. Furthermore, to avoid generating overly short and musically impractical segments, we enforce a minimum scene length of 8 seconds. This configuration allows us to partition long videos into a sequence of meaningful, temporally substantial clips suitable for individual soundtracking. It is noted that our video segmentation module is replaceable with some neural-based method such as AutoShot Zhu et al. (2023b). We choose PySceneDetect as a trade-off between speed and quality.

## B.3  VMPT TRAINING DETAILS

Our Visual-to-Music Prompt Translator (VMPT) is a Qwen3-8B model Yang et al. (2025). For parameter-efficient adaptation, we employed LoRA Hu et al. (2022) with a rank of 16, an alpha of 32, and a dropout of 0.1, targeting all linear layers. The model was trained for 3 epochs on our curated dataset, which contains approximately 17.8k pairs of video captions (input) and structured music captions (output).

The training was conducted using the LLaMA-Factory framework Zheng et al. (2024) on two NVIDIA A800 GPUs. We used the AdamW optimizer with a cosine learning rate schedule, a peak learning rate of 5e-5, a warmup ratio of 0.1, and a weight decay of 0.01. The training was performed with an effective global batch size of 16 and a maximum sequence length of 1024. To ensure training efficiency, we utilized bf16 mixed-precision and enabled gradient checkpointing. During inference, the trained LoRA adapter is merged into the base model, and text is generated using a sampling temperature of 0.7 and a maximum length of 512.

## B.4  VMPT PROMPT EXAMPLES

The VMPT is fine-tuned using a structured prompt designed to teach the model how to translate raw video captions into musical metadata. Figure 3 illustrates the complete prompt structure and an example used in our training.

```
### SYSTEM INSTRUCTION ###
You are a professional music analyst, you need to analysis the
given video caption and extract the main theme of the video, then
you need to find the music genre, instrument, and key that is most
suitable for the video.  Finally, you need to output the best music
caption that is most suitable for the video, do not need to output
the video theme, just output the music caption with the music genre,
instrument, and key.

### EXAMPLE INPUT ###
{A slow-motion montage shows a runner warming up at dawn on a quiet
city riverside.  Golden sunlight reflects on the water; close-ups
of tying shoelaces, deep breaths, and steady footfalls.  The camera
alternates between wide skyline shots and handheld tracking of the
runner's pace building from calm to determined.  Traffic is minimal,
birds are audible, and the mood transitions from introspective to
motivated as the runner starts the first sprint.}

### EXAMPLE OUTPUT ###
{Genre:  rock, pop, alternative rock, indie rock, Key:  F# major, BPM:
120, Instruments:  bass, drums, guitar, electric guitar, synthesizer,
Mood:  relaxing, happy}
```

Figure 3: The prompt structure and an input-output example used for fine-tuning the VMPT. The system instruction guides the LLM to act as a music analyst, and the example demonstrates how to convert a narrative video description into a structured music prompt with specific metadata.

### B.5 LLM AGENT PROMPT

The LLM Agent's decision-making is guided by a carefully constructed few-shot prompt. The prompt provides the model with a clear role, a description of the available tools, and several examples to demonstrate the desired reasoning process. Figure 4 shows the complete prompt structure.

## C EXPERIMENT SUPPLEMENTS

### C.1 LVS BENCHMARK DETAILS

A fundamental challenge in evaluating video soundtracking is the absence of a definitive ground-truth. Unlike objective tasks with a single correct answer, video soundtracking is an inherently subjective creative endeavor where a single video can be appropriately scored with a multitude of valid musical interpretations. Therefore, we posit that no single "ground-truth" soundtrack exists, and that fidelity to any original music is an inappropriate metric for quality. Guided by this principle, the curation of the LVS Benchmark focused not on finding videos with "ground-truth" music, but on selecting videos whose visual content provides a rich and challenging canvas for generative models. From an initial pool of over 1,000 candidates from the MMTrailer Chi et al. (2024) dataset, we manually selected 150 videos based on three primary criteria:

1. **Narrative Richness and Emotional Variation:** We selected videos that exhibit a clear narrative structure or a distinct emotional progression.

2. **Salient and Diverse Scene Transitions:** The benchmark prioritizes videos with a high density of clear and visually distinct scene changes.

3. **Rich Musical Potential:** We chose videos that offer clear, non-verbal cues for musical interpretation but are not rigidly tied to pre-existing diegetic music.

The final set consists of 120 video clips, each with a highly consistent duration of approximately 90 seconds (mean=90.02s, std=0.05s). The benchmark contains 567 segments in total, with an average of 4.72 segments per video. The core of the benchmark is its rich, structured annotations, generated via an automated pipeline, with an example shown in Figure 5.

```
### SYSTEM INSTRUCTION ###
You are a professional music director for video production.  Your
task is to select the most appropriate musical transition between
two consecutive video clips, Clip A and Clip B. Analyze the visual
and musical descriptions provided for both clips to determine the
narrative relationship between them.

Your available transition options are:
- generative transition:  For smooth, evolving changes in mood or
scenery.  Synthesizes a new musical bridge.
- abrupt cut:  For sudden, dramatic shifts in action or emotion.
- silent gap:  To create a moment of tension, surprise, or
reflection.
- fade-out/fade-in:  For simple, gentle changes between scenes with
similar moods.
You must only output the chosen transition style in lowercase.

### EXAMPLES ###

## Example 1
Clip A - Video Caption:  A wide, static shot of a serene lake at
dawn, with mist rising from the water.
Clip A - Music Caption:  Peaceful, ambient, slow strings, ethereal
pads.
Clip B - Video Caption:  A fast-paced car chase through a bustling
city at night, with quick cuts and explosions.
Clip B - Music Caption:  Action, intense, percussive, fast tempo,
heavy drums, cinematic hits.
Decision:  abrupt cut

## Example 2
Clip A - Video Caption:  A time-lapse of a single flower blooming in
the morning sun.
Clip A - Music Caption:  Hopeful, delicate, gentle piano melody, soft
strings.
Clip B - Video Caption:  A sweeping aerial drone shot revealing a
vast, lush green forest canopy.
Clip B - Music Caption:  Majestic, grand, orchestral, cinematic, full
strings section.
Decision:  generative transition

## Example 3
Clip A - Video Caption:  A character cautiously approaches a
mysterious, ancient door.
Clip A - Music Caption:  Suspenseful, low drone, tense, minimalist
synth.
Clip B - Video Caption:  The character's face, showing a look of pure
shock and disbelief after opening the door.
Clip B - Music Caption:  A sudden, loud, dramatic orchestral stab,
followed by silence.
Decision:  silent gap

### TASK ###
Clip A - Video Caption:  {Video Caption of Clip A}
Clip A - Music Caption:  {Music Caption of Clip A}
Clip B - Video Caption:  {Video Caption of Clip B}
Clip B - Music Caption:  {Music Caption of Clip B}
Decision:
```

Figure 4: The full few-shot prompt used to guide the LLM Agent.

```
{
  "video_id":  "_KI9GREleIw_82_122",
  "video_path":  ".../_KI9GREleIw_82_122.mp4",
  "video_caption":  "A group of four people are seated on a couch...",
  "video_duration":  90.0,
  "scenes":  [
    {
      "scene_index":  0,
      "scene_path":  ".../_KI9GREleIw_82_122_scene_000.mp4",
      "scene_duration":  13.56,
      "scene_caption":  "The video opens with a medium shot of four
people..."
    },
    { ... }
  ]
}
```

Figure 5: An example of the annotation structure for a single video in the LVS Benchmark. Each entry contains metadata and a caption for the full video, along with a structured list of its constituent scenes, each with its own metadata and caption.

## C.2 BASELINE IMPLEMENTATION DETAILS

To ensure a fair and rigorous comparison, we adapted all baseline models using their official open-source implementations and model checkpoints.

**CMT.** As this model generates music in MIDI format, we converted the output to waveform audio using the officially recommended synthesizer FluidSynth. To comprehensively evaluate its capabilities, we tested it in two settings. For the short-clip version ($CMT_S$), we generated music for each segment independently and concatenated the results using an abrupt cut. For the long-form version ($CMT_L$), we provided the entire 90-second video sequence as a single input to generate one continuous soundtrack.

**LORIS.** For LORIS, we used its 'dance_25s' checkpoint. Due to the model's limitations in generating long audio sequences, we only evaluated it in the segment-wise setting. Music was generated for each video segment individually and then connected via an abrupt cut.

**VidMuse.** Since VidMuse supports both long and short video inputs, we evaluated it in two settings analogous to CMT. For $VidMuse_S$, we generated music for each segment and concatenated the clips with an abrupt cut. For $VidMuse_L$, we generated a single continuous soundtrack by processing the entire long-form video at once.

**AudioX.** The default configuration of AudioX is to generate 10-second audio clips. Therefore, we only evaluated it in the segment-wise setting. For video segments longer than 10 seconds, we generated multiple consecutive 10-second audio clips to match the required duration and then concatenated them. All clips were connected using an abrupt cut.

