# OpenReview forum: "JenBridge: Adaptive Long-Form Video Soundtracking across Scene Transition"
_ICLR.cc/2026/Conference — ICLR 2026 Conference Withdrawn Submission_

### Official Review · Reviewer_ikiE · 2025-10-24

**Soundness:** 2
**Presentation:** 3
**Contribution:** 2
**Rating:** 4
**Confidence:** 5

**Summary:**

This paper presents JenBridge, a framework for generating long-form video soundtracks. It segments video, generates music per segment using a video-aware MMDIT, and introduces an adaptive transition mechanism. This mechanism uses an LLM Agent to select the optimal transition style to connect segments across scene changes. It introduce a comprehensive benchmark with rich annotations and a holistic evaluation protocol for long-form video soundtracking.

**Strengths:**

1. The work tackles video-to-music temporal alignment from a novel perspective: segmenting the video, generating music per scene, and then adaptively transitioning between these segments to form a complete soundtrack .
2. The proposed JenBridge framework offers a modular and interpretable approach to the complex task of long-form video soundtracking.
3. The paper introduces the LVS Benchmark, a comprehensive resource featuring rich annotations and a holistic evaluation protocol specifically designed for long-form video soundtracking .

**Weaknesses:**

1. The supplementary materials appear incomplete, lacking full comparative results for all examples and providing insufficient evidence to substantiate claims of coherence and fidelity across diverse transitions (only 3 examples).
2. Applying transitions only at segment boundaries may disrupt overall musical integrity, with examples showing potentially unnatural abrupt shifts.
3. Comparisons with recent SOTA methods like Diff-BGM [4], MuMu-LLaMA [1], GVMGen [2], and CCCG [3] are missing.
4. Evaluation is confined to the proposed LVS benchmark, lacking validation on other public datasets (e.g., BGM909 [4], SymMV [5]).
5. Key quantitative metrics such as FAD and KL divergence are absent from the objective evaluation.

[1]  MuMu-LLaMA: Multi-modal Music Understanding and Generation via Large Language Models

[2] GVMGen: A General Video-to-Music Generation Model with Hierarchical Attentions

[3] Customized Condition Controllable Generation for Video Soundtrack

[4] Diff-BGM: A Diffusion Model for Video Background Music Generation

[5] Video Background Music Generation: Dataset, Method and Evaluation

**Questions:**

1. Considering the variability inherent in generative music models (dependence on random seeds), how does the segment-wise generation and stitching methodology maintain global musical coherence across potentially disparate, independently generated segments, not just ensuring smooth local transitions?
2. Who performed the manual selection for the LVS benchmark curation ? Did this involve professionals with audio-visual production expertise?
3. Are all conditioning inputs (sequence text, global text, visual features) necessary ? Could ablations demonstrate their individual contributions?
4. Why were some metrics omitted from the ablation study results in Table 2 ?

---

### Official Review · Reviewer_Z7vX · 2025-10-26

**Soundness:** 2
**Presentation:** 3
**Contribution:** 2
**Rating:** 4
**Confidence:** 4

**Summary:**

This paper presents JenBridge, a framework for adaptive audio transitions between video-clips that enables high-quality long-form video soundtrack generation. JenBridge consists of a video segmentation module, a foundational text-to-music model trained on a private dataset, and an adaptive transition module orchestrated by a LLM agent that considers both visual captions and musical prompts. Generative transition technique leverages ControlNet-based inpainting model, interpolating between adjacent clips using slerp on text embeddings and latent representations. Additionally, the authors propose LVS Benchmark, constructed from videos filtered from the MMTrail dataset, accompanied by scene boundary annotations and visual description generated by captioning model. JenBridge showed state-of-the-art generated soundtrack quality on both objective and subjective evaluations.

**Strengths:**

1. JenBridge can generate long-form soundtrack for videos of arbitrary length using its adaptive transition module.

2. JenBridge outperforms baseline models in long-from soundtrack generation on both video-music alignment metric and musical aesthetics metric.

3. The authors successfully fine-tuned LLM/VLMs for V2M task and integrated them into JenBridge.

**Weaknesses:**

1. In the introduction, the authors point out that previous V2M pipelines were typically unmodifiable, giving users less creative control on intermediate steps or the final output. They then state JenBridge tries to address this limitation. However, it is still unclear in what sense the users can creatively modify or control the intermediate phase within the JenBridge framework.

2. In the abstract, the authors state that LVS Benchmark contains novel metrics, but all the objective metrics in the benchmark were preexisting ones. (ImageBind Score, Meta Audiobox Aesthetics)

3. The Transition Naturalness metric in user study for CMT_s is 1.6 and VidMuse_s is 1.8. In ablation studies, the transition metric for JenBridge with only Abrupt Cut is 2.8. Why do the authors think JenBridge without adaptive transition outperforms CMT_s and VidMuse_s, even thought they all use same transition methods? In my opinion, if Transition Naturalness is a metric for evaluating the transition smoothness and appropriateness as explained by the authors, evaluated results for these models should not differ very much.

4. An ablation study of dropping both Adaptive Transition and Visual Conditioning would better demonstrate the effectiveness of core contributions of JenBridge, since the private text-to-music model alone already seems to improve performance a lot.

**Questions:**

Many details are missing in the paper, hindering full understanding of the proposed approach and its experiments. For examples, the following questions should be elaborated:

1. What are the ratios of selected transition types?
2. Why did the authors interpolate between latent representations L_{k, i} and L_{k+1, N-i+1}, not L_{k, i} and L_{k+1, i} or other choice of pairs?
3. How many videos were evaluated by participants in the user study?
4. How did the metrics omitted in Table 2 compared to those in Table 1 change in the ablation study?
5. Are the durations of transitions fixed? What are the durations for each transition?
6. It would be nice to provide a demo of generated results by JenBridge, compared on other baselines.

---

### Official Review · Reviewer_KL2U · 2025-10-30

**Soundness:** 3
**Presentation:** 3
**Contribution:** 2
**Rating:** 2
**Confidence:** 5

**Summary:**

This paper addresses the challenge of generating coherent, long-form musical soundtracks for videos with dynamic scene changes. The authors argue that existing video-to-music models are primarily designed for short, isolated clips and lack mechanisms for ensuring narrative continuity across transitions.

To solve this, they propose JenBridge, a modular framework that operates in a three-stage process. First, it segments a long video into a sequence of semantically coherent clips. Second, a video-aware diffusion model generates a distinct piece of music for each individual clip. The core contribution lies in the third stage: an adaptive transition mechanism. This mechanism employs a LLM Agent, acting as a "director," to intelligently select the most appropriate transition style from a versatile toolkit to connect the music segments.

To facilitate evaluation, the paper also introduces the Long-form Video Soundtrack (LVS) Benchmark, a benchmark specifically designed for this task. The authors claim that on this benchmark, JenBridge significantly outperforms existing methods in both objective and subjective metrics, particularly in transition naturalness and overall narrative coherence.

**Strengths:**

*   **Well-Motivated Problem Formulation**: The paper is motivated by a significant and practical challenge in video-to-music generation: creating coherent, long-form soundtracks that adapt across dynamic scene transitions. This provides a clear and relevant goal for the work.

*   **A New Approach for Music Transition**: The paper introduces an adaptive transition mechanism directed by an LLM Agent. The concept of using an LLM to select from a toolkit of transition styles based on narrative context is a distinct contribution. This method introduces a layer of semantic reasoning to the process of stitching music segments.

*   **Modular and Interpretable Architecture**: The JenBridge framework is designed in a modular fashion (segmentation, generation, transition). This architectural choice makes the system interpretable and allows for potential control over intermediate steps, which aligns with the stated goal of creating a practical tool.

**Weaknesses:**

While the motivation is interesting, the paper's contribution is undermined by severe weaknesses in its evaluation, methodology, and overall academic rigor.

1.  **Insufficient Experimental Evaluation**

    The paper's performance claims are unsubstantiated due to an incomplete experimental setup.

    *   **Missing Standard Metrics:** The evaluation omits crucial, community-standard audio quality metrics like Fréchet Audio Distance (FAD). This absence makes a direct and objective comparison of the model's audio fidelity against prior work difficult and leaves the assessment of quality incomplete..
    *   **Limited Generalization on Established Benchmarks:** The method is not evaluated on existing long-form video-to-music benchmarks (e.g., **SymMV**, **V2M-bench**), raising questions about its generalization capabilities. Evaluating solely on LVS prevents a clear understanding of how the model performs relative to the state of the art in established settings.
    *   **Confounded Baseline Comparison:** The comparison is uninformative as it conflates the contribution of the proposed transition logic with the underlying generator's performance. A proper ablation would apply the same transition logic to baselines to disentangle these effects.

2.  **Weaknesses in Methodological Justification**

    The paper primarily describes a system's architecture without providing sufficient scientific justification or empirical validation for its core design choices.

    *   **Unmotivated Architectural Complexity:** The choice of a complex, two-stage VLM-LLM pipeline over a more direct, single end-to-end model (e.g., Qwen-VL) is not adequately justified. The architecture appears ad-hoc and is not supported by ablation studies that demonstrate the superiority of this specific, more complex configuration over simpler alternatives.
    *   **Outdated Components:** The use of a 2022 EnCodec model is a significant limitation, as modern, higher-fidelity codecs could substantially alter the results.
    *   **Unfulfilled Premise:** The paper claims to address the "unmodifiable" nature of prior models but fails to demonstrate how its complex pipeline offers any practical improvement in user controllability.

3.  **Poorly Motivated Benchmark**

    The necessity of the new LVS benchmark is not established. The authors do not differentiate it from existing benchmarks like **SymMV** or explain what unique scientific questions it addresses.

4.  **Lack of Academic Rigor**

    *   **Citation Errors:** The paper contains frequent and noticeable citation errors.
    *   **Report-Style, Not Research:** The overall presentation lacks the analytical depth, rigorous ablations, and generalizable insights expected of a research paper, reading instead like a system description.

**Questions:**

1.  **On Evaluation Rigor**: The evaluation lacks standard audio quality metrics (like FAD/FD) and is not benchmarked on any existing long-form video datasets (like SymMV or V2M-bench). Can you provide these results to properly situate your work and validate its performance?

2.  **On the LVS Benchmark's Contribution**: Could you clarify the necessity of the LVS benchmark over existing long-form benchmarks? What unique aspects does it evaluate that prior benchmarks do not?

3.  **On Architectural Justification**: What is the motivation for using a complex VLM-to-LLM pipeline instead of a single, more direct vision-language model (e.g., Qwen-VL)? Can you provide an ablation or rationale to support this specific design choice?

4.  **On Isolating the Transition Module's Contribution**: To provide a fairer assessment and isolate your core contribution, could you apply your proposed LLM-directed transition mechanism to the outputs of a strong baseline generator like AudioX and report the results?

---

### Official Review · Reviewer_Jcim · 2025-10-31

**Soundness:** 3
**Presentation:** 3
**Contribution:** 2
**Rating:** 6
**Confidence:** 4

**Summary:**

This work presents a video sound-tracking generation pipeline for long videos across scene transitions. The music generation model has a two-stage design: a pretrained text-to-music generation model, and a visual-to-music prompt translator based on a Qwen3 LLM to translate a video caption to a music prompt. It also categorizes transition types into four categories and either uses a fixed audio transition or a controlNet model to add the music clip during the transition.  The transition type is determined by a LLM agent based on Qwen3.

**Strengths:**

-	The proposed two-stage design is novel. The translator design can convert a powerful text-to-music backbone into a video-to-music generator.
-	Experimental results are strong. The proposed method outperforms existing methods AudioX and VidMuse on both subjective metrics and user preference.
-	The ablation study is comprehensive.

**Weaknesses:**

-	One drawback of the two-stage method is that it loses some critical information for background music generation. For example, the motion of objects is not included as cues. The model cannot generate music piece that matches the movement patterns of the person/objects in the video. Authors should discuss more on this potential issue.
-	The design of transition types is kind of ad hoc. It is more preferred to have a unified model to predict the transition types and produce the music at the same time.

**Questions:**

na

---

### Note · Authors · 2025-11-13

I have read and agree with the venue's withdrawal policy on behalf of myself and my co-authors.